# CLESP: Collaborative Learning with Ensemble Soft Pseudo-Labels

## Abstract

In this work, we present Collaborative Learning with Ensemble Soft Pseudo-Labels (CLESP), a method for updating a set of pre-trained classifiers on unlabeled data during the testing phase. CLESP improves the classification performance of ensembles by allowing member classifiers to learn from each other and improve during inference. Specifically, we minimize the cross-entropy between the classifier soft output that has the highest predicted probability for the majority-voted class (a high confidence/entropy softmax) and all other classifiers. The majority-voted model that all others learn from may change from sample to sample, allowing the group to dynamically learn from each other without labels. Our method uniquely optimizes all trainable parameters in each model and applies to both single-sample settings and batch settings. In our experiments, using sets of independently pre-trained base classifiers with distinct architectures, we find that CLESP can significantly reduce generalization errors of ensemble models on classification tasks such as the CIFAR-10, CIFAR-100, and ImageNet datasets and their corrupted counterparts, while also minimizing the entropy of classifier soft outputs.

## 1 Introduction

Modern classifiers can reliably produce accurate predictions when training and testing data are drawn from identical distributions. When there is a *distribution shift* between the training (source) and testing (target) data, i.e. the sets of source and target samples are drawn from distinct but related distributions, then the recognition performance of many high variance classifiers significantly diminishes (Quionero-Candela et al., 2009; Liang et al., 2023). *Covariate shifts*, which are distribution shifts along only the input feature distribution, are quite common in many real-world ML problems, such as when sensors in medical imaging equipment degrade enough to distort readings (Karani et al., 2021; Hu et al., 2024), or when self-driving cars encounter unique weather conditions not included in the original training data (Li et al., 2025). Methods belonging to Domain Adaptation (Singhal et al., 2023) address such cases where model deployment may still be necessary even if off-the-shelf pretrained models cannot achieve low error on distribution-shifted data.

In this work, we propose a new psuedo-labeling method to self-train ensembles of pretrained classifiers on testing data in image classification settings. We show that minimization of our proposed loss signal reduces ensemble diversity, counter to common ensemble heuristics that encourage diversity among member models (Buschjäger et al., 2020), and we empirically verify that `CLESP` can adapt sufficiently-competent pretrained models to new distribution shifts. We provide a simple ensemble combination function that utilizes information from pretrained member models to produce psuedo-labels for self-training. When minimizing the standard cross-entropy loss between member outputs and the ensemble psuedo-label, we find that larger reductions in member errors occur on datasets with larger distribution shifts. We verify our claims using the CIFAR10, CIFAR100, and ImageNet datasets, as well as their corrupted counterparts (Hendrycks & Dietterich, 2019) which exemplify covariate shifts.

**Contributions:**

- **Dynamic Psuedo-Labeling:** We introduce a simple ensemble combiner that produces soft psuedo-labels from pre-trained model outputs.

- **Adaptation:** Surprisingly, `CLESP` framework achieves its largest improvements on distribution shifted data as opposed to in-distribution data.

- **Ensemble Collaboration:** `CLESP` allows ensemble members to dynamically learn from each other. Diverse ensembles will have the greatest potential to improve, and diversity can be artificially inflated by adapting to more severe distribution shifts. Additionally, `CLESP` can significantly reduce labeling costs, as we show that `CLESP` can improve classification accuracies without labels.

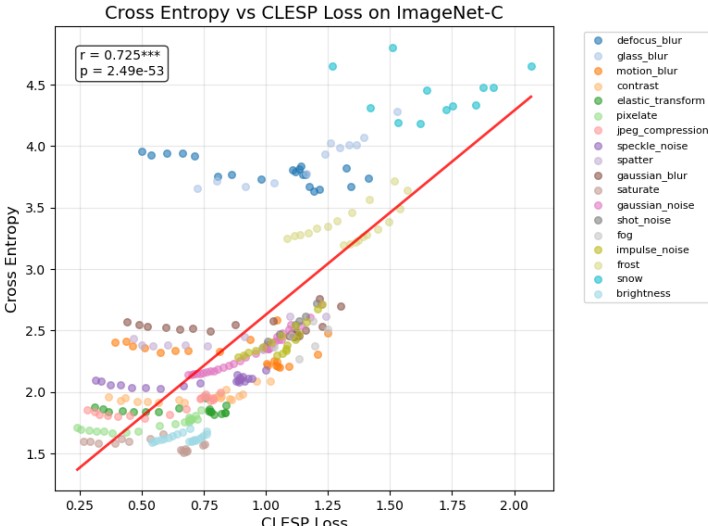

Figure 1: Cross-Entropy of Ensemble Output $H(\mathbf{x})$ with Ground-Truth $\mathbf{y}$ vs CLESP Loss during adaptation on various ImageNet-C corruption types. We observe a high Pearson correlation coefficient along with highly significant p-values. The Spearman Rank Correlation is $\rho = 0.7452$ , which along with the reported Pearson correlation coefficient shows a strong positive correlation between the standard ground-truth cross-entropy and our `CLESP` loss.

## 2 RELATED WORKS

### 2.1 DOMAIN ADAPTATION & TEST-TIME ADAPTATION (DA & TTA)

The foremost aim in Domain Adaptation and Test-Time Adaptation is to improve the performance of a source pre-trained model on a label-scarce target domain when the source and target are not identically distributed (Zhao et al., 2023). DA methods assume access to labeled source training data during inference, whereas TTA methods tend to allow access to only the unlabeled test data. Many lines of work in TTA have found success with minimizing the Shannon Entropy of a model's softmax output as a signal in the absence of labels (Gui et al., 2024; Zhang et al., 2022). Recently, several new studies have shown that reducing only the prediction entropy can result in problematic model behaviors, such as leading models to produce over confident predictions (poor calibration) and inducing catastrophic forgetting when spurious correlation shifts are encountered (Lee et al., 2024; Yoon et al., 2024). *Catastrophic forgetting* is defined as erroneously overwriting model weights by learning from incorrect psuedo-labels. Our proposed signal optimizes several other signals of interest by proxy (Figure 1, Figure 2) as opposed to only minimizing softmax entropy.

Methods like TENT (Wang et al., 2021) are able to mitigate these difficulties by re-estimating batch statistics on new data (as opposed to optimizing all trainable parameters), but not all TTA use cases allow access to more than a single sample at a time. In the setting where only a single sample is

available at a time, EATA proposed to use weight-averaged and augmentation-averaged predictions for entropy optimization, while ROID explicitly depends on forming an ensemble from prior versions of an adapted model to facilitate generalization to new domains (Marsden et al., 2023; Niu et al., 2022). All of these methods are able to adapt a single model at a time to new domains, while we choose to focus on the simultaneous adaptation of multiple models to new data through the ensemble-combined soft psuedo-label. Although our method minimizes the Shannon Entropy of model predictions as a side effect (Figure 2, left), we rely on the inherent diversity in a set of independently pre-trained models to make our psuedo-labels more accurate (Figure 2, right).

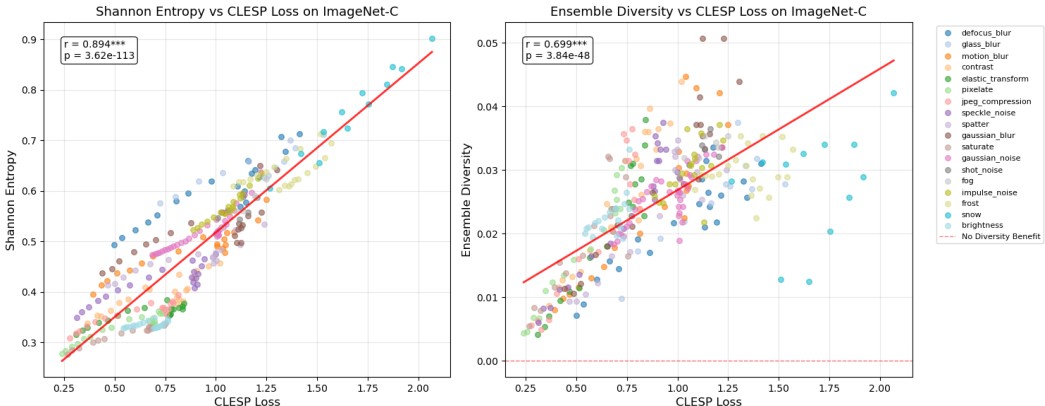

Figure 2: Average Shannon Entropy of Ensemble Members vs CLESP Loss during adaptation on various ImageNet-C corruption types (left), Ensemble Diversity vs CLESP Loss during adaptation on various ImageNet-C corruption types (right). In both plots we observe a high Pearson correlation coefficient (denoted by r) and highly significant p-values. The Spearman Rank Correlation for the Shannon entropy plot is $\rho = 0.8801$, and for the Diversity plot is $\rho = 0.7292$, which in tandem with the reported Pearson correlation coefficients show a strong positive correlation in both cases.

COTTA avoids catastrophic by stochastically restoring small subsets of the model weights to their original values during each iteration, whereas we make no attempts to avoid catastrophic forgetting (Wang et al., 2022). We aim to show the viability of true ensemble based approaches in TTA settings using member models that have been independently pre-trained. The Shannon Entropy can be interpreted as measuring the uncertainty present in a softmax layer, so entropy minimization approaches effectively minimize the uncertainty in model predictions.

## 2.2 ENSEMBLE LEARNING

Ensemble learning combines the outputs of multiple member models to generate more accurate predictions than any of the constituent models alone (Dietterich, 2000). Popular methods include bagging, boosting, stacking, voting, and Bayesian model averaging (Breiman, 1996; Freund & Schapire, 1997; Schapire & Singer, 1999). Since the objective in the standard Ensemble Learning setting is to only achieve better predictions, constituent models remain unchanged.

A simple way to combine the predictions of models on an arbitrary input sample is to choose the class that was most frequently predicted by the member models, a process referred to as Majority Voting. It is well known that ensembles with high diversity (made from models which make independent errors) have improved prediction robustness, where the individual model errors cancel each other out during the model combination process (Wang & Ji, 2023). Despite the common use of ensemble learning in modern machine learning methods, the presence of diversity among ensemble members is often thought of as a "black box" for improving classification accuracy. Still, many works on ensembles indicate that to get a strong ensemble, diversity should be present among members.

We can view the cross-entropy between member models and the majority voted model as a rudimentary estimate of the diversity in an ensemble, and by minimizing it we force members to agree with the majority voted model. Recent work has shown that the diversity of an Ensemble directly relates to the degree of error reduction that the ensemble provides over its member models (Wood et al., 2023). A natural counter-example to this claim is to form an ensemble from member models that are

each an identical copy of one perfect model and its pre-trained weights such that the copied model achieves 100% classification accuracy on validation data. Such an ensemble would have no diversity among its identical members, but the overall ensemble would retain perfect accuracy, perhaps contingent upon the choice of model combination method used. As such, we expect that is possible to reduce diversity while also reducing generalization error in an ensemble.

## 2.3 KNOWLEDGE DISTILLATION

Knowledge Distillation uses a complex teacher model to supervise a smaller student model. Transferring knowledge to a smaller model reduces computational overheads and allows well-performing models to be deployed to devices with limited resources, like mobile phones and embedded systems. One early line of work found success in training a small student model on test data with hard-targets generated by a pre-trained ensemble (Bucila et al., 2006). However, using one-hot encodings as pseudo-targets effectively forces the student model to never improve beyond the static teacher since it can draw no additional knowledge about the zeroed out classes in the one-hot target.

(Hinton et al., 2015) found that using a teacher model's real-valued softmax output with a temperature parameter $T > 1$ can provide more information rich pseudo-targets which better facilitated generalization in student models as opposed to using generated "hard-targets", that is, one-hot pseudo-targets predicted by the teacher. This follows the idea that given an input sample $x$ that can belong to one of $k$ classes and a teacher model $H : \mathcal{X} \mapsto \mathbb{R}^k$ , the probabilities of the incorrect classes relative to each other in the teacher's soft-max contain relevant information about similarities between classes. In our work we opt to have ensembles produce softmaxes to take advantage of their higher self-information content compared to one-hot pseudo-labels, but do not employ any temperature control.

However, in Knowledge Distillation teacher and student models are usually independent, and learning is one-way and affects only the student model and not the teacher (Gou et al., 2021). Work following this line of thought has found its best results when the teacher outputs used in the distillation process came from the same input data that the teacher was initially trained on. We propose letting the teacher model be an ensemble that is dependent on the output of the student, so that the distillation process becomes dynamic for both the student and the teacher.

## 3 METHODOLOGY

### 3.1 SET-UP

For an input space $\mathcal{X}^{train}$ and $k$-class label space $\mathcal{Y} = \{c_1, c_2, ..., c_k\}$, let $h_\theta : \mathcal{X} \mapsto \mathcal{Y}$ denote a classifier with weights $\theta$ pretrained on source training data $\mathcal{D}^{train} = \{(\mathbf{x}_j^{train}, c_j)\}_{j=1}^N \subset \mathcal{X}^{train} \times \mathcal{Y}$. Assume that we have a set of $m$ such classifiers each pretrained on $\mathcal{D}^{train}$ and parameterized by weights $\theta_i$ for $i = 1, ..., m$. We do not require that the ensemble classifiers have identical architectures, only that members output softmax vectors corresponding to the $k$ classes in $\mathcal{Y}$, so unless otherwise specified we will assume that any two classifiers with distinct weights also have distinct architectures. Our goal is to further train pre-trained ensembles on unlabeled testing data $\mathcal{D}^{test} = \{(\mathbf{x}_j^{test}, c_j)\} \subset \mathcal{X}^{test} \times \mathcal{Y}$, assuming $\mathcal{X}^{train} \neq \mathcal{X}^{test}$, by using the pre-trained knowledge of each classifier.

Denote an ensemble combined output of all $k$ member models on an input sample $\mathbf{x}$ as $H(\mathbf{x})$. A simple and popular way to combine the outputs of ensemble members is to take the majority vote over their output classifications Dogan & Birant (2019). On input $\mathbf{x}$, assume that the majority of models predict class $c \in \mathcal{Y}$. We define our proposed ensemble output as the softmax output of the member model which has the highest predicted probability for the majority voted class $c$,

$$H(\mathbf{x}) := \underset{\{h_{\theta_i}(\mathbf{x})|i=1,...,M\}}{\arg\max} h_{\theta_i}(\mathbf{x})^{(c)}. \tag{1}$$

Additionally, $H$ is the psuedo-label that we propose all other models in the ensemble learn from. An advantage of $H$ as defined above is that $H(\mathbf{x})$ will agree with the standard majority vote classification while also providing softmax information about classes other than $c$. Additionally, since

probability distributions with high mass concentrations tend to have low entropy, the ensemble member that is given by $H(\mathbf{x})$ (i.e. member with high probability concentration on majority voted class) will also tend to have low Shannon entropy. The minimization of the Shannon entropy of classifier soft outputs has been a standard practice in Test-Time and Domain Adaptation problems ((Wang et al., 2021), (Liang et al., 2023)). By avoiding a one-hot encoding of the majority-voted class our ensemble psuedo-label: 1) must be supported by multiple member models, and 2) will carry softmax information about other classes' relative similarities to the voted class ((Hinton et al., 2015), (Gou et al., 2021)).

## 3.2 Collaborative Learning with Ensemble Soft Pseudo-Labels

For CLESP we propose to minimize the summed cross-entropy between all model outputs and the soft target $H(\mathbf{x})$, with our loss defined as,

$$\mathcal{L}(h_{\theta_1}, h_{\theta_2}, ..., h_{\theta_n}, \mathbf{x}) = \frac{1}{M} \sum_{i=1}^{M} \text{CE}(h_{\theta_i}(\mathbf{x}), H(\mathbf{x})), \tag{2}$$

where CE is the standard cross-entropy function. One advantage of using the majority voted softmax for the ensemble output is that the model deemed the strongest by the group should be least affected by backpropagation using the above loss. If the $j$-th model is selected as the output, so $H(\mathbf{x}) = h_{\theta_j}(\mathbf{x})$, then the $j$-th term of the sum in (3) is the cross-entropy of $h_{\theta_j}(\mathbf{x})$ with itself. By the definition of cross-entropy, $\text{CE}(h_{\theta_j}(\mathbf{x}), h_{\theta_j}(\mathbf{x}))$ is simply the self-entropy of $h_{\theta_j}(\mathbf{x})$, and since $h_{\theta_j}(\mathbf{x})$ was selected to have low self-entropy it follows that it will make only small contributions to the loss. It is worth noting that by using the cross-entropy in this way we risk overwriting the strong knowledge of the majority voted models, as we do not explicitly address catastrophic forgetting in our approach. The use of a statistical distance function that is 0 when inputs are identical, like the Kullback–Leibler divergence, will be investigated in future work.

## 3.3 Loss Decomposition & the Role of Ensemble Diversity

Following Wood et al. (2023), we can consider the 0/1 loss (0 if both inputs are the same, 1 otherwise) in expectation over all data as a simple measure of ensemble classification error. Assuming that $\mathbb{E}_D[\ell_{0/1}(\mathbf{h}_{\theta_i}, \mathbf{y})] \approx \mathbb{P}[\mathbf{h}_{\theta_i} \neq \mathbf{y}]$, we can reveal terms in the error that account for base classifier errors and diversity,

$$\ell_{0/1}(H, \mathbf{y}) = \frac{1}{m} \sum_{i=1}^{m} \ell_{0/1}(h_{\theta_i}, \mathbf{y}) - \frac{1}{m} \sum_{i=1}^{m} \left( \ell_{0/1}(h_{\theta_i}, \mathbf{y}) - \ell_{0/1}(H, \mathbf{y}) \right)$$

$$\underbrace{\mathbb{E}_D[\ell_{0/1}(H, \mathbf{y})]}_{\text{Expected Ensemble Error}} = \underbrace{\frac{1}{m} \sum_{i=1}^{m} \mathbb{P}[h_{\theta_i} \neq \mathbf{y}]}_{\text{Average Member Error}} - \underbrace{\frac{1}{m} \sum_{i=1}^{m} \left( \mathbb{P}[h_{\theta_i} \neq \mathbf{y}] - \mathbb{P}[H \neq \mathbf{y}] \right)}_{\text{Ensemble Diversity}} \tag{3}$$

The first term is the average base classifier error and the second term is the average difference between the ensemble error and the base classifier errors, representing the base classifiers' diversity about a well-chosen ensemble combiner. Clearly, a complete minimization of the CLESP loss in (2) will reduce the diversity term present in (3) (see figure 2, right). Assuming that CLESP minimizes ensemble diversity, then as we might expect, (3) implies that the average error of ensemble members must increase more than the diversity decreases.

By using the majority vote combiner to produce psuedo-labels with greater accuracy than the average base classifier, we aim to reduce the average error while also minimizing the ensemble diversity. When the majority vote combiner is consistently incorrect, we find that diversity decreases more than average model errors, resulting in *catastrophic forgetting*. Additionally, we use a small class-balanced and labeled portion of incoming test data as a form of validation set to determine optimal termination points during CLESP . If the average model accuracy on the validation set degrades beyond an acceptable range, we reject the current model weights and return either 1) the last version of the member model weights that showed improvements in average classification accuracy, or 2) the

initial pre-trained models when performance degradation is present from the first epoch of `CLESP`.

## 3.4 ALGORITHM

The iterative adaptation procedure for `CLESP` is illustrated in Algorithm 1, and consists of the following 3 phases:

**Initialization** The optimizer collects all trainable parameters from all $M$ pretrained models used for CLESP. We require a small labeled validation set sampled from the test set and for all models to have a pretrained accuracy over 50%. Additionally, we freeze all batch normalization layer weights so that they are not updated during adaptation.

**Iteration** For each batch of data, the `CLESP` loss is backpropagated to all models. If we have a sample for which there is no well-defined majority voted class (e.g. all classes receive the same amount of votes), then we discard that sample to avoid risking error propagation from uncertain psuedo-labels.

**Termination** Before applying CLESP, users must set aside a class-balanced portion of their unlabeled testing data and hand-label the samples so that they can be used as a validation set. We assume that the labeled validation set and the unlabeled data to adapt to will be from the same distribution. It follows then that the accuracy or ground-truth loss improvements that we achieve on the labeled data by adapting to the unlabeled data should generalize to other unlabeled samples. We show a correlation analysis between the ground-truth loss and the validation loss in Appendix A. In cases where models immediately degrade on the first epoch we will simply return the initial pre-trained models with weights untouched by CLESP.

After every iteration, the accuracy of each member model is estimated in the labeled validation set, and if the classification accuracy (or loss) shows consistent degradation over a range of iterations, then the algorithm is terminated. Empirically, ensembles that started `CLESP` with below 50% accuracy on the validation set were found to degrade much more consistently. The trials that showed the largest ensemble accuracy improvements were found to have degraded only after 100 epochs. Additionally, we multiply our loss by a very small regularization constant $10^{-11}$, which we touch more on in appendix B.

---

**Algorithm 1** Intra-model Ensemble Learning (CLESP)

---

**Require:** Set of $M$ pretrained source models $S = \{h_{\theta_1}, h_{\theta_2}, ..., h_{\theta_M}\}$, set of unlabeled test-set samples $\mathcal{X}^{test}$ with $k$ classes for adaptation, set of labeled test-set samples for validation.
    On input $x \in \mathcal{X}^{test}$, determine majority voted class $c$
    $H \leftarrow \arg\max_{\{h_{\theta_i}(\mathbf{x})|i=1,...,M\}} h_{\theta_i}(\mathbf{x})^{(c)}$         ▷ Set ensemble output
    $\mathcal{L} = \frac{1}{M} \sum_{i=1}^{M} \text{CE}(h_{\theta_i}(\mathbf{x}), H(\mathbf{x}))$         ▷ Compute loss
    Back propagate $\mathcal{L}$ to all models in $S$.
    Estimate accuracy on validation set.
    If validation accuracy is too low, terminate algorithm and return the models with their current weights.

---

## 4 EXPERIMENTS

We evaluate the capability of `CLESP` to adapt a set of pre-trained classifiers to test set data in both a standard setting (where test and train data are drawn from the same distribution) and a covariate shift setting (where input the feature distribution is shifted). We adapt a freshly instantiated ensemble of pre-trained classifiers to each corruption type in each dataset (uncorrupted data has only the single corruption type 'uncorrupted'), and all models were implemented with PyTorch (Paszke et al., 2019) with the adaptation process utilizing up to 2 NVIDIA L40S GPUs.

**Datasets**
For experiments in the standard setting we use samples from the validation sets provided by the

CIFAR10, CIFAR100 and ImageNet datasets. In the covariate shift setting we similarly use samples from the validation sets provided by the CIFAR10-C, CIFAR100-C, and ImageNet-C datasets. This ensures that no samples used during the pre-training of member models are seen again during adaptation, no matter if the samples were corrupted or uncorrupted. Each corrupted dataset contains 15 corruption types (e.g. Gaussian noise, JPEG compression noise, etc.), with each corruption type having 5 distinct severity levels. We utilize corrupted samples with only the maximum severity level 5.

In each experiment, we adapt a freshly instantiated pre-trained ensemble to 70% of available data from a single corruption type without using labels. The remaining 30% of available data from the same corruption type is split into a validation set used to determine a termination point and an evaluation set used to estimate generalization error. In the covariate shift setting, once adaptation on a particular corruption type is finished, we instantiate a new pre-trained ensemble and repeat the process on the next corruption type.

**Models** Experiments ran on a particular dataset and its corrupted counterpart used identical base models for adaptation, so in the corrupted case the members were still pre-trained on uncorrupted samples. This leads to poorer initial accuracies on corrupted data, which also induces higher diversity in the ensemble, and thus a large CLESP loss to optimize. Additionally, as adaptive batch normalization layers have been successfully used for adaptation tasks in the past, we make the decision to freeze the batch normalization layers of all models in all experiments to avoid conflating the effects of our method with the effects of adaptive batch norm.

For CIFAR10 and its corrupted counterpart, as well as CIFAR100 and its corrupted counterpart, the initial ensemble was formed with base models of the following architectures: ResNet-20 (He et al., 2015), VGG-11 (Simonyan & Zisserman, 2015), MobileNetV2 (Sandler et al., 2019), ShuffleNetV2 (Ma et al., 2018), and RepVGG (Ding et al., 2021). For CIFAR10/CIFAR10C experiments the base models were trained on uncorrupted CIFAR10, and in CIFAR100/CIFAR100C experiments the base models were trained on CIFAR100. All model weights for both CIFAR10 and CIFAR100 were sourced from a public GitHub repository whose owners we have no affiliations or relations with. For ImageNet and its corrupted counterpart, the initial ensemble was formed with base models of the following architectures: ResNext101-64x4 (Xie et al., 2017), ResNet152, ResNet101, and ResNet50. All model weights for ImageNet were sourced from the PyTorch Vision library.

## 4.1 EXPERIMENT RESULTS, ANALYSIS AND DISCUSSION

Below in Figure 3 we show the mean accuracy improvements that CLESP achieves after termination, with confidence intervals provided by a hierarchical bootstrap across corruption types with 10,000 resamples. CLESP was ran at least twice on each each corruption type (including uncorrupted), giving us at least two sets of results for each corruption type. Since experimental results that come from the same corruption type and/or same dataset are not necessarily independent, we first resampled corruptions from each dataset with replacement, and then for each corruption we resampled again among the results available for that corruption type and dataset.

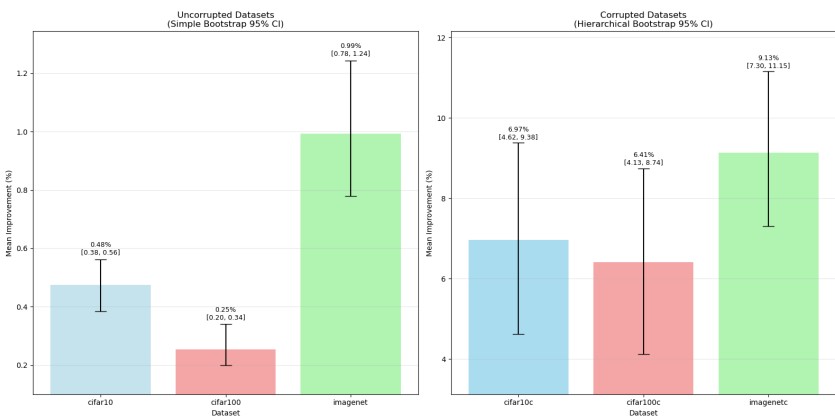

Figure 3: Mean improvement in average model accuracy during adaptation on each dataset.

With 10,000 resamples we found that `CLESP` could increase the average member accuracy on the evaluation set quite consistently, albeit only by 1% or less (Figure 3, left). On corrupted data we find much higher improvements in classification accuracy, above 5% on all corrupted data sets, with the lower bounds of improvements on each dataset still being positive (Figure 3, right). One possible explanation is that pre-trained member models have much higher initial accuracies on testing data from the same distribution as their source training data, leading to less diversity in the models' predictions on uncorrupted data.

In Figure 4 we can see the average accuracy of member models vs epochs during adaptation on the CIFAR10-C fog corruption. All plots show clear signs of improvement in classification performance and do not yet show signs of overfitting, even at 500 epochs. This also demonstrates the long-run behavior of `CLESP`, as we can see that at high epoch counts we still get moderately stable training curves.

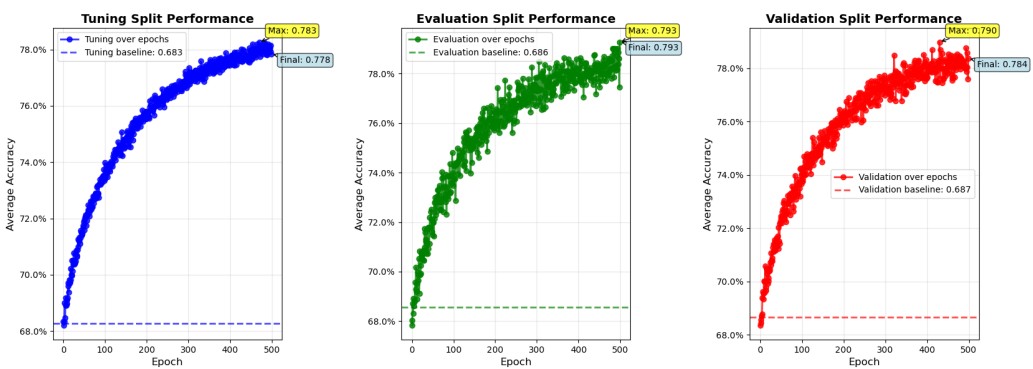

Figure 4: Average Member Accuracy vs Epochs during adaptation on the frost corruption type from CIFAR10-C. The tuning split refers to data used for adaptation, the evaluation split refers to data used strictly for estimating generalization error, and the validation split refers to data used strictly for determining a termination point.

## 5 CONCLUSION

**`CLESP` can improve generalization error of pre-trained models.** Figure 3 show thats `CLESP` successfully increases classification accuracy on both corrupted and uncorrupted data, demonstrating `CLESP`'s ability to adapt to covariate shifts. Additionally, for some corruption types we observe that the increase in classification accuracy is higher on the unseen evaluation set data than on the tuning set data, showing that `CLESP` can avoid overfitting (Figure 4).

**`CLESP` enables models to learn from each other.** Since our optimization objective minimizes the cross-entropy between models in the ensemble, all performance gains observed stem directly from within the ensemble, showing true intra-model ensemble learning. This shows a significant difference from conventional ensemble learning methods that leave individual models unchanged.

**`CLESP` reduces entropy and error without dependence on batch normalization layers.** Figures 1 and 3 show that the Shannon Entropy of individual model outputs, the ensemble diversity, and the ground-truth cross-entropy loss are all highly correlated with our proposed `CLESP` loss.

**Limitations and Future Work** Similar to many standard Ensemble Learning methods, `CLESP` is more computationally heavy than TTA methods that employ only one model and do not use backpropagation. However, we believe that it is a cost worth paying as `CLESP` is able to affect all trainable parameters in models and adapt multiple models simultaneously. It would be interesting to analyze the behavior of `CLESP` as we change the number of models in our set, as many works in Ensemble Learning indicate that having larger ensembles can significantly improve the resulting reduction in generalization error of the ensemble output compared to the outputs of the individual models. Additionally, there are many techniques for avoiding catastrophic forgetting that were not employed in `CLESP`, but could be considered in future work, such as replacing the cross-entropy

with a statistical distance function that outputs 0 when both input distributions are identical, or stochastically restoring subsets of model weights after backpropagation.

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

# A  TERMINATION CONDITION

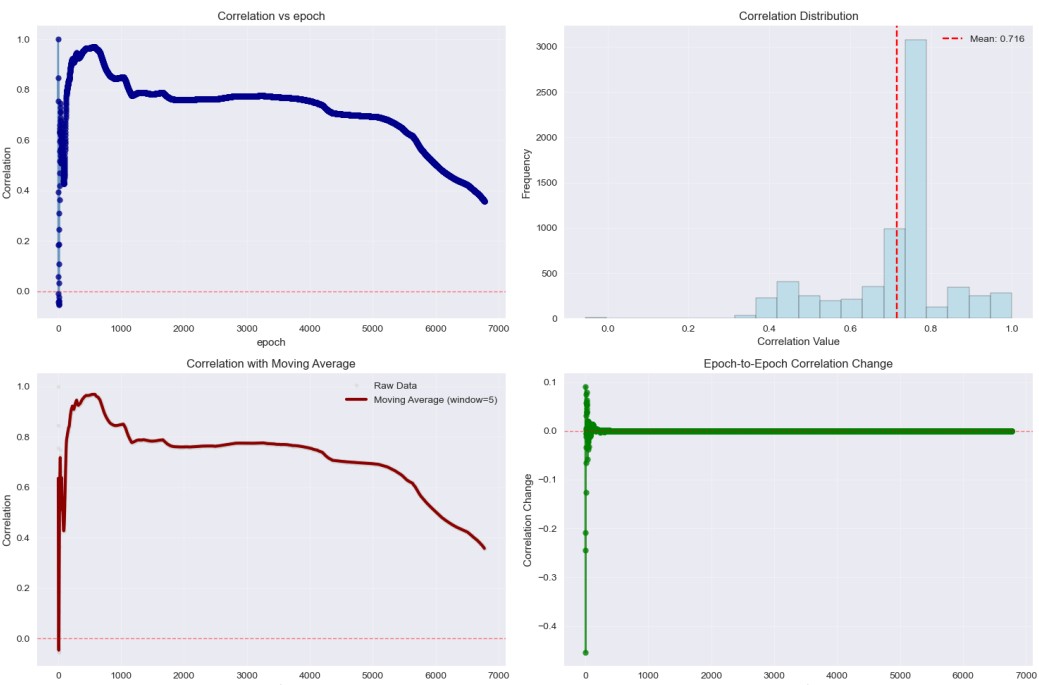

Figure 5: Correlation between validation set and evaluation set losses during adaptation to a corruption type in CIFAR10-C.

Above we provide correlation plots between the evaluation set loss and validation set loss during adaptation. Correlation provides a way to measure how closely changes in evaluation performance align with validation performance, which offers insight into the stability and reliability of training beyond accuracy alone. We tested four different data splits for tuning, validation, and evaluation (70/20/10, 75/15/10, 80/10/10, and 85/5/10). Using multiple validation splits helps avoid relying on a single partition and shows how well the model generalizes. Tracking correlations under these conditions also provides a useful factor for determining when to stop training. Once correlations stop rising or begin to decrease, it suggests that further training is unlikely to improve generalization, although the evaluation set loss is typically not available during adaptation.

Tracking the correlation between evaluation loss and validation loss is useful for seeing whether improvements during training lead to better generalization. When the correlation is high and steady, then drops in evaluation loss will match drops in validation loss, showing that the model is learning patterns that transfer beyond the training data. If the correlation falls, it suggests the model is starting to overfit, since improvements in evaluation loss are no longer similar to validation performance. This means correlation provides an additional check beyond loss values for monitoring training progress and determining when to stop before overfitting.

## B  LOSS LANDSCAPE

The weight space of the ensembles spans the collective weights of all member models, leading to a very high-dimensional loss surface. To encourage stable convergence and avoid needing a more comprehensive analysis of the complex `CLESP` loss landscape, we use an extremely small learning rate ($\alpha = 10^{-11}$) during our experiments.

