# OpenReview forum: "CLESP: Collaborative Learning with Ensemble Soft Pseudo-Labels"
_ICLR.cc/2026/Conference — ICLR 2026 Conference Withdrawn Submission_

### Official Review · Reviewer_sh1S · 2025-10-28

**Soundness:** 2
**Presentation:** 1
**Contribution:** 2
**Rating:** 2
**Confidence:** 4

**Summary:**

This paper introduces a collaborative test-time adaptation framework for ensemble classifiers that addresses performance degradation under distribution shift by enabling models to learn from one another through soft pseudo-labels.
The proposed approach dynamically selects the most confident ensemble member as a teacher for each sample and aligns the remaining members via cross-entropy, allowing adaptation without access to ground-truth labels.
The algorithmic contribution is validated through extensive experiments on CIFAR-10, CIFAR-100, and ImageNet (including their corrupted variants), demonstrating consistent accuracy improvements in classification tasks.

**Strengths:**

It presents a new idea on ensemble classifiers on distribution shifts. However, it has some weakness in completeness and novelty.

**Weaknesses:**

(1)
The paper suffers from significant issues in writing quality, logical flow, structure, and formatting. A paper with a clear core contribution and strong experimental evidence can still be convincing even with a shorter page count. However, in this case, the submission lacks substantive contributions, while allocating an excessive portion of space (approximately two pages) to the Related Work section, which gives the impression that the authors are compensating for the absence of meaningful technical content.

In addition, numerous concrete errors are found throughout the paper, including but not limited to

	-The word “pseudo” is misspelled as “psuedo” across multiple sections (Introduction, Related Work, Methodology)

	-Formatting inconsistencies (e.g., Figure 1 caption, inconsistent boldface or structure in the Experiments section’s Datasets / Models subsections)

	-Mathematical notation (e.g., line 206: “all k members” → should be “all m members”)

	-Incorrect equation reference (e.g., “j-th term of the sum in (3)” refers to the wrong equation)

	-Inconsistent variable usage (mixing m and M for the number of models)

	-Equation (1) does not clearly define h(x)as the class probability output of each model

	-Equation (3) mixes E[⋅]and P[⋅]even after stating they can be treated equivalently, leading to mathematical inconsistency

	-Terminology inconsistencies (e.g., abbreviations not handled properly, “Domain Adaptation” vs “DA,” “Shannon entropy” vs simply “entropy”)

	Mismatch between main text and figure descriptions (e.g., fog vs frost in Figure 4)

These issues collectively weaken the clarity and credibility of the paper and significantly hinder readability.


(2)
The proposed method is overly simplistic and remains at the level of combining components that are already well-known in the literature. The authors themselves acknowledge that using soft pseudo-labels to avoid information loss inherent in one-hot encoding is a widely established technique, which further diminishes the novelty of the approach.
A more fundamental issue is that the authors explicitly admit that their loss function can lead to catastrophic forgetting, yet they present the method without offering any mechanism to address this problem. This significantly undermines the legitimacy of the proposed approach. If the paper had instead provided a solution to this failure mode, that alone could have constituted a meaningful contribution.
The paper also implies that the diversity-related term contributes to greater stability. However, this rationale conflicts with the fundamental advantage of ensemble learning, where diversity among models is precisely what enables robustness and improved reliability. Actively reducing diversity risks negating the core benefit of using an ensemble in the first place. A more compelling direction would have been to preserve diversity while preventing catastrophic forgetting, which would have led to a far more convincing and impactful contribution.

(3)
The experimental validation is insufficient to support the proposed methodology. The paper does not provide convincing evidence for when and why CLESP is effective, nor does it demonstrate whether the method yields meaningful improvement compared to existing approaches.

Figure 1 appears to suggest that the CLESP loss decreases in the same direction as the ground-truth loss even without labels. However, the paper does not describe the experimental setup, dataset configuration, or measurement process, and it does not even reference or explain Figure 1 in the main text. Furthermore, simply showing a correlation is not persuasive enough to establish the superiority of the proposed loss function.

Figure 2 is essentially used as supporting evidence for the proposed method, yet it is placed in the Related Work section, which makes the narrative feel disjointed. If this figure is intended to justify the core logic of the approach, it should be clearly explained within the Methodology section instead.

Figure 3 appears to be the main experiment of the paper, but it only reports the difference in accuracy before and after applying CLESP. Showing “a 1% improvement” or “a 5% improvement” is not sufficient to judge the effectiveness of the method, especially when no comparison to baseline TTA techniques is provided. Without such baselines, it is impossible to determine whether CLESP is genuinely strong or merely providing marginal gains.

Additionally, although the paper focuses on image classification task, it provides no qualitative examples illustrating how CLESP operates. There is no visualization showing how ensemble members initially behaved, how the majority pseudo-label was formed, or how this led to improved evaluation performance. As a result, the reader cannot intuitively understand the adaptation behavior or verify that CLESP works as intended.

**Questions:**

I have mentioned problems and questions in the "WEAKNESS" sections

**Details Of Ethics Concerns:**

It does not have ethical concerns

---

### Official Review · Reviewer_MKry · 2025-10-29

**Soundness:** 3
**Presentation:** 3
**Contribution:** 2
**Rating:** 2
**Confidence:** 4

**Summary:**

The paper studies ensembling of pre-trained classifiers during test phase. It proposes CLESP, Collaborative Learning with  Ensemble Soft PSeudo-Labels to improve the performance of the ensemble by letting individual classifiers learn from each other. The classifier with the best prediction is chosen and the cross-entropy loss of others is minimized with respect to its soft prediction. Experiments on cifar10/100/imagenet and their corrupted versions show that CLESP improve the generalization of the ensemble.

**Strengths:**

- The paper is well-written and the approach is intuitive, simple yet efficient
- The problem tackled is of great interest and the approach is original and versatile with the possibility to consider different types of models in the ensemble

**Weaknesses:**

I list below what I believe are weaknesses but I would be happy to be corrected if I misunderstood some parts of the work.

- The experiments are not convincing with poor alignmenet (correlation and spearman coefficient with low values aroung 0.7/0.8 do not qualify for a strong alignement, see Fig. 1 and 2 (right)).
- The approach requires to have access to a few labeled test data which seems unrealistic (and unfair to other approaches ) in the scenario of pseudo-labeling with unlabeled data.
- There is no baseline to compare with CLESP. In particular, the traditional pseudo-labelling and ensembling methods should be used as baselines. One notable approach from [1] uses unlabeled data to force diversity with a loss term similar to Eq. 3). Since in the current submission, diversity is found to decrease more than the model errors, which is opposite to what was found in [1, 2], it would be interesting to discuss those works.
- This is not a weakness per se but visualization should be improved to better see the legends and numerical values

Overall, while the approach is interesting, the practical implementation is unconvincing with the requirement to have access to some labeled test data and unclear improvement in terms of performance. The current submission lacks proper comparison to common baselines to showcase the benefits of the approach. In its current state, I lean towards rejection for these reasons.

*References*

[1] Odonnat et al. Leveraging Ensemble Diversity for Robust Self-Training in the Presence of Sample Selection Bias. AISTATS 2024

[2] Zhang et al. Exploiting unlabeled data to enhance ensemble diversity. Data Mining and Knowledge Discovery 2013

**Questions:**

Could the atuhors add baseline comparisons, for instance, pseudo-labeling and common ensemble methods such as majority vote?  [1] would also be an interesting comparions (although it does not require pre-trained individual classifiers).

---

### Official Review · Reviewer_Ap8U · 2025-10-31

**Soundness:** 1
**Presentation:** 2
**Contribution:** 2
**Rating:** 2
**Confidence:** 4

**Summary:**

This paper proposes the CLESP method, which enables collaborative learning among multiple pre-trained models during the testing phase by integrating soft pseudo labels. The method dynamically selects the model output with the highest prediction probability within the majority voting class as the pseudo label, minimizing the cross-entropy between other models and this pseudo label. It achieves improvements on distribution of shifted data as opposed to in-distribution data. And experiments on several datasets demonstrate that CLESP significantly reduces generalization error while decreasing classifier output entropy. It is applicable to both single-sample and batch scenarios.

**Strengths:**

This work constructs a framework of multiple pre-trained models ensemble to improve the problem of covariate shifts between source domain data and target data, and the proposed method adapts sufficiently-competent pretrained models to new distribution shifts.

**Weaknesses:**

1. This work lacks a thorough analysis of the research motivation, fails to provide an in-depth examination of the shortcomings in existing studies, and also omits a comparative analysis with prior work (including similarities, differences, and advantages).

2. This paper utilizes ensemble learning to mitigate distribution shifts between training and testing datasets. By utilizing multiple pre-trained networks, it designs an ensemble strategy for joint learning and optimization, including the formulation of a joint learning objective function. However, the designed function draws upon existing concepts within ensemble learning. It is recommended that authors follow a major module of ensemble learning: how to combine multiple classifiers, where many researchers focus on designing a better strategy of ensemble all classifiers to improve the performance. Actually, the strategy proposed by this work is not particularly innovative. Additionally, in an ensemble system, each learner is always named as an ensemble individual, not member, ‘member’ is not professional.

3. One concept may be incorrectly understood by authors. Actually, in ensemble learning, two key factors must be considered:  individual error and diversity among individuals. An ideal ensemble should be composed of individuals with lower error and higher diversity, rather than lower diversity. Hence, in this ensemble of multiple pre-trained models, how to enhance the diversity while reducing the error of individuals. This is also the tough point for ensemble learning, since higher diversity is incompatible with lower individual error.

4. Furthermore, when multiple pre-trained models are employed to address a given problem, the resultant computation complexity is notably elevated. The question therefore arises as to how this issue should be addressed.

5. The presentation of this paper is also unclear. It has been observed that many descriptions of mathematical symbols are lacking in precision and consistency, such as m and M. It is suggested that the writing be revised so that this work can be better followed.

6. In experiments, the comparison of the proposed method to other existing methods is also scarce. Add to some recent works related to ensemble strategies and domain adaptations/generalization.

**Questions:**

See the comments on weaknesses.

---

### Official Review · Reviewer_sBpw · 2025-11-01

**Soundness:** 2
**Presentation:** 2
**Contribution:** 1
**Rating:** 4
**Confidence:** 3

**Summary:**

The paper proposes a simple ensemble test-time adaptation rule. On each input, they pick the majority-voted class, then select the member with the highest probability on that class to serve as a soft teacher. They train all members by minimizing cross-entropy to the teacher. Furthermore, they show the superior performance of their method in the covariant shift task on multiple datasets.

**Strengths:**

* The method is simple and easy to apply, and it shows improvements under covariate shift.
* Correlation analyses (Figs. 1–2) suggest CLESP loss tracks entropy, diversity, and ground-truth CE, which is interesting.

**Weaknesses:**

* The novelty of the paper is limited. The main idea of the paper resembles known mutual learning and self-distillation ideas, where models learn from each other’s predictions (e.g., Deep Mutual Learning, co-teaching, born-again networks). The paper cites knowledge distillation broadly but does not position CLESP against mutual and collaborative distillation or ensemble-teacher baselines that use the averaged softmax as a teacher.
* The evaluation details are not specified (e.g., optimizer, seeds).
* There is no comparison with other test-time adaptation methods, or to simpler ensemble teachers like mean-softmax or hard majority pseudo-labels.
* The method requires hand-labeling a subset of target data to choose the stopping point, contradicting the “no labels” narrative and making the setting closer to semi-supervised target adaptation.

Minor weakness:

* Numerous typos (e.g., psuedo-label appears throughout), grammar errors (line 412).

**Questions:**

* Why is the majority-class, max-confidence single-member teacher preferable to the mean softmax teacher?
* How large is the labeled validation set (absolute count per dataset)? What happens with no labels?

---

### Note · Authors · 2025-12-03

I have read and agree with the venue's withdrawal policy on behalf of myself and my co-authors.